# Associations between Body Appreciation and Disordered Eating in a Large Sample of Adolescents

**DOI:** 10.3390/nu12030752

**Published:** 2020-03-12

**Authors:** Migle Baceviciene, Rasa Jankauskiene

**Affiliations:** 1Department of Physical and Social Education, Lithuanian Sports University, Sporto 6, 44221 Kaunas, Lithuania; 2Institute of Sport Science and Innovations, Lithuanian Sports University, Sporto 6, 44221 Kaunas, Lithuania; rasa.jankauskiene@lsu.lt

**Keywords:** body appreciation, disordered eating, self-esteem, self-objectification, sports, adolescents

## Abstract

Body appreciation is one of the main facets of a positive body image. The present study aimed to examine the psychometric properties of the Lithuanian version of the Body Appreciation Scale-2 (BAS-2-LT) and test the associations between body appreciation and disordered eating in a large sample of adolescents of both genders. **Method:** The sample consisted of 1412 adolescents (40.2% were boys). The ages ranged from 15 to 18 years (92.4% were 17), with a mean age of 16.9 (SD = 0.5) for girls and 17.0 (SD = 0.4) for boys. Participants completed the BAS-2-LT alongside the measures of body dissatisfaction, disordered eating, body mass index, self-esteem, body functionality, and participation in sports. Linear regressions were used to test the associations between study variables and disordered eating. **Results:** BAS-2-LT replicated the original one-dimensional structure in girls and boys. Invariance across genders was established. The instrument showed good internal consistency and temporal stability. Body appreciation was negatively correlated with higher levels of body mass index, body dissatisfaction, and disordered eating. Positive associations were observed between body appreciation, self-esteem, body functionality, and sports participation. Higher levels of body appreciation decreased the risk of disordered eating behaviors in both genders. **Conclusions:** The results of the present study support the psychometric properties of BAS-2-LT. Body appreciation is associated with lower disordered eating in adolescent girls and boys. These findings present empirical support for the development of interventions programs that promote positive body images and aim to prevent disordered eating in adolescent boys and girls.

## 1. Introduction

Body image research has traditionally been focused on the exploration of negative body image and its predictors [1]. Negative facets of body image such as body dissatisfaction, a drive for thinness, thin ideal internalization, overweight preoccupation, and body shame were analyzed, concluding that negative body image is associated with disordered eating, poorer psychological and physical health, and obesity [2]. However, research has demonstrated that positive rather than negative body image has more consequences for various life domains [3]. Nevertheless, for decades, research tended to conceptualize positive body image as one endpoint along a body image continuum, with positive body image anchored at the opposite endpoint to negative body image [4]. However, in the last two decades, the study of positive body image has demonstrated that it is a construct that is distinct from negative body image and is multifaceted, holistic, protective, and adjustable via interventions [5]. The central facet of the positive body image is body appreciation, which is defined as accepting, holding favorable opinions toward, respecting the body, resisting the pressure to internalize stereotyped beauty standards as the only form of human beauty, and appreciating the functionality and health of the body [5]. Qualitative studies in adolescent girls have demonstrated that despite endorsing some appearance concerns and social appearance comparison, adolescent girls with positive body image express the appreciation of body image differences, self-confidence, body–self connection, self-attunement, and empowerment and report strategies that help to mitigate the potential negative association between social media exposure and body image [6,7]. Longitudinal study in adolescent girls demonstrated that body appreciation prospectively predicted a decrease in dieting, alcohol, and cigarette use, and an increase in physical activity by one year of observation [8].

The majority of studies exploring positive body image measured body appreciation and used the Body Appreciation Scale, a 13-item, one-dimensional measure [9]. This scale measures three components of body appreciation: acceptance of the body regardless of size or perceived imperfections; respect and care of the body using healthy self-care behaviors; and protection of the body through resistance to internalize stereotyped beauty ideals presented by media [9]. The scale demonstrated good test–retest stability, internal consistency, convergent validity with body image and disordered eating-related measures, as well as measurement equivalence/invariance between genders in U.S. samples [5,10]. However, the structure of the scale as a one-dimensional construct was not replicated in several non-English-speaking samples [11,12]. Additional limitations of the scale were the development of positive body image research ignoring item content, low factor loadings of some items, and different wordings for one item for women and men [4].

The second version of the Body Appreciation Scale (BAS-2) was developed to overcome its previous limitations [4]. BAS-2 has 10 items, with five items being new and five belonging to the original instrument. The BAS-2 has been validated in numerous samples of different countries and languages, i.e., Malaysia [13], Greece [14], United Arab Emirates [15], Poland [16], Romania [17], Dutch speaking students in the Netherlands [18], the Netherlands [18], Brazil [19], China [20], Hong Kong [12], Iran [21], Portugal [22], Denmark, Portugal and Sweden [23], Brasilia [24], and Mexico, Argentina, and Columbia [25]. Fortunately, BAS-2 showed a one-dimensional factor structure in all samples of different cultures. Furthermore, the scale demonstrated good test–retest stability [4] and good internal consistency (α = 0.87 to α = 0.96) in the majority of studies.

The BAS-2 demonstrated convergent validity with appearance evaluation [4,14,16,18], body pride [15], self-esteem [4,14,20,23], proactive coping [4], psychological well-being [16,19,23], life satisfaction [13,20], quality of life [14], and intuitive eating [4,23]. Negative associations were observed between the BAS-2 scale and body dissatisfaction, internalization of the thin ideal [4], social physique anxiety [24], self-objectification [18], and disordered eating behaviors [4,19]. Some studies reported negative associations between BAS-2 and body mass index [14,16,18,20]. The positive body image is associated with the appreciation of the functionality of the body [5]. Some studies have demonstrated that student athletes have reported higher body appreciation compared with non-athletes [26].

Some differences emerged in the levels of body appreciation for women and men. For example, some studies observed no gender differences [13,20], while others demonstrated lower BAS-2 scores in women than in men [4,19,21,23,24,25]. Furthermore, the majority of studies reported measurement invariance by gender. Invariance between genders was established in the U.S. sample [4], adolescents and young adults in Danish, Portuguese, and Swedish samples [23], and Brazilian [19], Polish [16], Chinese [20], and Portuguese [22] adult samples. However, it was not supported in the Romanian sample [17] and only partially supported in adolescents from Brazil and Denmark [23,24].

Understanding the protective nature of positive body image, it is important to more deeply investigate the associations between body appreciation and health-related lifestyles, especially in adolescence. Adolescence is a period of dramatic psychological and physical changes, with body image playing one of the most important roles [27]. The development of a positive body image is understudied in adolescent samples worldwide. Furthermore, only several studies validated BAS-2 in adolescent samples. In particular, BAS-2 was validated in Danish, Portuguese, and Swedish [23], Brazilian [24], and Mexican, Argentinian, and Columbian samples of adolescents [25]. To the best our knowledge, there have been no BAS-2 validation studies in Eastern European countries. A significant proportion of Lithuanian adolescents report body image concerns and health-compromising eating behaviors [28]. The promotion of a positive body image is one of the essential purposes of disordered eating prevention and healthy lifestyle promotion-related programs [29]. The validation of the BAS-2 will facilitate future research into the positive body image of young people in Lithuania.

The majority of studies were implemented in young adults, demonstrating negative associations between body appreciation and disordered eating [4,19] The research in adult women demonstrated that disordered eating is associated with inflexible eating rules and that body appreciation is a mediator of the associations between social safeness and more flexible eating rules [30]. Unfortunately, the associations between positive body image and eating behaviors are understudied in adolescent populations. Therefore, the present study will add to the knowledge on this issue.

The aim of the present study was to examine the psychometric properties of the Lithuanian version of the BAS-2 in a sample of adolescent females and males. Firstly, we expected that the Lithuanian version of the Body Appreciation Scale-2 (BAS-2-LT) would replicate the original unidimensional structure. In addition to focusing on understudied adolescent populations, this study aimed to assess the measurement invariance between genders expecting to establish it. Second, we aimed to evaluate the convergent validity of the instrument using measures of body mass index (BMI), body dissatisfaction, disordered eating behaviors, self-esteem, functionality of the body, and participation in sports. Based on the previous studies, we expected that boys would score higher than girls on the BAS-2-LT and that the BAS-2-LT scores would be negatively associated with BMI, body dissatisfaction, and disordered eating, but positively associated with self-esteem, body functionality, and participation in sport in adolescent girls and boys. Finally, we aimed to explore the associations of body appreciation with disordered eating in adolescent girls and boys, expecting that body appreciation would be associated with significantly lower levels of disordered eating in both genders when controlling for other study variables.

## 2. Materials and Methods 

The sample consisted of 1412 adolescents (40.2% of them were boys) from 26 Lithuanian cities and 41 randomly selected gymnasiums representing municipalities of the largest country cities. Participants were from the 11^th^ grade. The ages ranged from 15 to 18 years (92.4% were 17), with a mean age of 16.9 (SD = 0.5) for girls and 17.0 (SD = 0.4) for boys. Self-reported BMI of the sample ranged from 14.0 to 41.7 kg/m^2^, while the mean BMI for girls was 21.0 (SD = 3.0) kg/m^2^ and for boys was 22.0 (SD = 3.1) kg/m^2^. To analyze the temporal stability of the BAS-2, forty-four volunteer students (24 girls and 20 boys) were invited to complete the same questionnaire two weeks after they had first completed surveys to investigate the test–retest reliability of the LT-BAS-2.

### 2.1. Procedure

This study is part of the larger study conducted in 2019. The study questionnaire consisted of four domains: lifestyles (physical activity, participation in sports, nutrition, sedentary behavior, sleep duration, smoking, and alcohol consumption), body image (BAS-2-LT, body dissatisfaction, drive for thinness, drive for muscularity, self-objectification, and sociocultural attitudes towards appearance–4 scales), disordered eating and dysfunctional exercise scales, and self-esteem and self-rated health. School directors and informed parent consents were obtained, providing permission for schoolchildren to participate in the study. Respondents provided their answers by filling in the online questionnaires consisting of a battery of self-report questionnaires designed to measure study variables. Overall, 1492 students participated in the study. However, 56 students refused to participate by themselves. Furthermore, 24 questionnaires were deleted as they were not completed correctly. For the final analysis, 1412 questionnaires were used containing no missing data.

### 2.2. Ethical Considerations

The researchers obtained ethical approval to conduct this study from the Committee for Social Sciences Research Ethics of the Lithuanian Sports University (protocol No. SMTEK-32, 27-09-2019). The students were provided a possibility to select option “I agree to participate” or “I disagree to participate” to give their consent to participate in the study before beginning the survey. Following the fundamental ethical and legal principles of the research, the students were introduced to the aim of the study before the questionnaires were provided. The laws of anonymity and goodwill were followed.

### 2.3. Measures 

#### 2.3.1. Body Appreciation

The 10-item Body Appreciation Scale-2 [4] assesses three facets of body appreciation (body acceptance, respect to one’s body, and resistance to pressure from media’s appearance ideals). The instrument comprises 10 items rated on a 5-point Likert-type scale (1 = Never, 5 = Always). The mean score of the scale was calculated by averaging all items, thus the scoring range possible to obtain varied from 1 to 5. Higher scores indicate greater body appreciation. The translation of the BAS-2 into Lithuanian was carefully performed by a professional translator and then back-translated to English by two professional translators from a translation agency in Kaunas, Lithuania. The original version and developed translation were reviewed by translators, and the final version of the translation was approved. The face validity was rated as good.

#### 2.3.2. Self-Esteem

Self-esteem was measured by M. Rosenberg’s Self-Esteem Scale (RSES) [31]. This scale measures individual global self-esteem. The scale consists of 10 items. Half of the items are positively worded, while the other half are negatively worded. The items are scored on a 4-point Likert scale, ranging from 1 (strongly disagree) to 4 (strongly agree). Negatively worded items are reversed and an overall self-esteem score is computed, yielding scores from 10 to 40. A higher score indicates a greater level of self-esteem. Cronbach’s alpha for the RSES in this study was 0.86.

#### 2.3.3. Body Dissatisfaction

Body dissatisfaction was assessed using the body dissatisfaction subscale from the Eating Disorder Inventory-3 (EDI-3) [32]. The body dissatisfaction subscale comprises 10 items with Likert-type answers from always (4) to never (0) with the greater values indicating higher body dissatisfaction. The body dissatisfaction subscale assesses discontentment with the size and shape of one’s body. The mean score of the scale was calculated by averaging all items, thus the scoring range possible to obtain varied from 0 to 4. The scales had adequate psychometric qualities in adolescent and young adults nonclinical samples [28,33,34]. In the present sample, the internal consistency of the body dissatisfaction subscale was Cronbach’s α = 0.82.

#### 2.3.4. Self-Objectification Questionnaire

Self-Objectification Questionnaire (SOQ) [35] assesses whether a person views his/her body in an objectified, appearance-related manner or in a nonobjectified, body functionality-based manner. Participants ranked 10 body attributes in order of importance to them. Five of the items are associated with appearance-based attributes (body weight, physical attractiveness and measurements, sexuality, firm/sculpted muscles), while the other five relate to body functionality-based physical attributes (health, strength, physical fitness, physical coordination, and energy level). Scores were calculated by assigning value to the rankings, with the most important receiving 10 and the least important attribute receiving 1. Furthermore, the appearance-oriented attribute rankings were added together for one total, and the body functionality-based ones for the second. Next, body functionality-based attributes were subtracted from appearance-based ones. Thus, the final score ranged between 25 and −25. The higher the score, the more the adolescent objectifies his/her body. In the present study, we used the body functionality subscale only.

#### 2.3.5. Disordered Eating

Disordered eating was assessed by the Eating Disorder Examination Questionnaire 6.0 (EDE-Q) [36]. EDE-Q comprehensively measures the essential behavioral characteristics of disordered eating behavior and/or eating disorders. EDE-Q consists of 28 items. The EDE-Q 6.0 assess the individual eating-related behaviors in the last 28 days. The first six open-ended questions measure the frequency of the essential behavioral characteristics of eating disorders, i.e., binge eating, self-induced vomiting, laxative use, and excessive exercise. The next 22 attitudinal questions form four subscales. The subscales’ scores reflect the severity of the disordered eating. The answer options are arranged on a 6-point Likert scale from 0 (no days) to 6 (every day). The mean score of the scale is calculated by averaging all items, thus the scoring range possible to obtain varies from 0 to 6. A higher score indicated either greater severity or frequency of disordered eating. The Lithuanian version of the scale demonstrated good psychometric properties [37]. Internal consistency for the general scale was good (α = 0.95).

#### 2.3.6. Body Mass Index

Body mass index (BMI) was calculated by self-reported height and body weight. Children were classified into four body mass categories: thin, normal weight, overweight, and obese using the extended international (IOTF) body mass index cut-offs for thinness, overweight, and obesity [38]. For schoolchildren ≥18 years, adult body mass index standards were used to define underweight (<18.5 kg/m^2^), normal weight (18.5–24.9 kg/m^2^), overweight (25.0–29.9 kg/m^2^), and obesity (≥30 kg/m^2^). It was found that 16.0% of boys and 10.6% of girls were either overweight or obese and 4.9% of boys and 16.5% of girls were underweight (*p* < 0.001).

#### 2.3.7. Participation in Sport

Participation in sport was assessed using two questions: “Do you participate in leisure sports?” and “Do you participate in competitive sports?” Participation in leisure and competitive sports was combined into one group. Overall, 83.5% of boys and 70.1% of girls declared participation in sports (*p* < 0.001).

### 2.4. Statistical Analysis

First, descriptive statistics of the sample were performed, the results of which are presented as the means, medians, standard deviations, and percentages of the minimum and maximum values scored as floor and ceiling. Skewness was calculated to indicate the degree of distortion from the symmetrical bell curve or the normal distribution, and Kurtosis as an indicator of heavy tails or outliers. Intraclass correlation coefficients (ICCs) were calculated for assessing test–retest reliability. Cronbach’s alpha coefficients were used for the evaluation of internal consistency. A score of ≥0.90 was considered as excellent [39]. Pearson’s correlation coefficients were used for the analyses of construct validity (inter-item correlations). Correlations of the value <0.40 were considered weak, 0.40–0.59 as moderate, and ≥0.6 as strong [39]. Third, to confirm the concurrent validity, Pearson’s correlation coefficients were used to evaluate the relationships between the BAS-2-LT scores and the measures from the life satisfaction score, RSES, Eating Disorder Inventory, Body Dissatisfaction (EDI-BD), Lithuanian Eating Disorder Examination Questionnaire-6 (LT-EDEQ-6.0), and BMI calculations. To test the predictive power of body appreciation, self-esteem, body dissatisfaction, body functionality, and body mass index on disordered eating behaviors, multiple linear regression analysis was performed. Next, the construct validity of the BAS-2-LT was studied by performing exploratory factor analysis (EFA), and then confirmatory factor analysis (CFA). The sample was randomly divided into two equal groups. One group was used for EFA (*n* = 706), and another split-half group for CFA (*n* = 706). The EFA was performed using principal component analysis extraction method with the rotation method of Varimax with Kaiser normalization. Then, using AMOS (analysis of momentary structure), the CFA of the 10-item BAS-LT-2 scale was conducted, and the goodness of fit of the model was assessed using acceptable fit values: the comparative fit index, CFI (0.90 < CFI < 0.95), and the root of the mean square error of approximation, (RMSEA) (0.05 < RMSEA < 0.08). Finally, structural invariance between gender groups of the BAS-2-LT was tested. The statistical analyses were carried out using IBM SPSS Statistics 26 (IBM Corp., Armonk, NY, USA) and AMOS version 24 (IBM Corp., Armonk, NY, USA).

## 3. Results

The descriptive statistics for the BAS-2-LT results are presented in Table 1. Median, mean, standard deviation, range, kurtosis, skewness, and percent scoring at the lowest possible value (floor) and the highest possible value (ceiling) were demonstrated to report the descriptive characteristics of the instrument. For girls, the global BAS-2-LT scale score was lower as compared with boys (3.24 ± 1.13 and 3.41 ± 1.16, respectively, *p* < 0.01). The skewness and kurtosis coefficients were computed for the data distribution normality analysis purposes. In boys and girls, the LT-BAS-2 scores were moderately negatively skewed.

In boys and girls, the Kaiser–Meyer–Olkin (KMO) resulted in a measure of sampling adequacy of 0.96, and Bartlett’s test of sphericity (for boys χ^2^ = 6438.8, df = 45, *p* < 0.001; for girls χ^2^ = 9411.4, df = 45, *p* < 0.001) indicated the appropriateness to proceed with exploratory factor analysis (Table 2). We used the Varimax method to obtain orthogonal factors. Using this method, a 5-factor solution was revealed. The one-factor model accounted for 78.1% of the total variance for boys and 78.4% for girls.

The one-factor structure identified via EFA was next evaluated through CFA (Figure 1). The standardized estimates of factor loadings were all adequate, but the initial CFA indicated a poor model fit (goodness of fit index (GFI) = 0.92; adjusted goodness of fit index (AGFI) = 0.88; Tucker Lewis index (TLI) = 0.96; comparative fit index (CFI) = 0.97; root of the mean square error of approximation (RMSEA) = 0.104, standardized root mean square residual (SRMR) = 0.037). Since fit indices were not found to be acceptable, modification was considered to improve model fit. Specifically, modification indices were consulted to free error covariance between items 1 and 2, 2 and 4, 3 and 4, 4 and 6, 2 and 9, and 9 and 10. These modifications resulted in an adequately fitting model: (GFI = 0.96; AGFI = 0.92; TLI = 0.97; CFI = 0.98; RMSEA = 0.083, SRMR = 0.029).

Next, we determined whether the BAS-2-LT was invariant between gender groups in the adolescent sample (Table 3). Invariance was tested by comparing the fully unconstrained model to models increasingly constrained in terms of factor loadings, structural covariance, and residual item variance. Invariance analyses between gender groups revealed a statistical difference between unconstrained and fully constrained models. The statistically significant differences were found when testing the assumption about measurement residual equalities, but not factor loadings and structural covariance between genders.

Table 4 represents the results of internal consistency and test–retest reliability of the BAS-2-LT scale separately in boys’ and girls’ subsamples. Findings confirmed good test–retest reliability and a Cronbach’s alpha for boys and girls of 0.97.

A comparison of scoring of the study variables is presented in Table 5. Boys demonstrated higher life satisfaction scores, whereas girls expressed higher body dissatisfaction and more frequent disordered eating behaviors. Body appreciation positively correlated with self-esteem and a higher ranking of body features representing body functionality in girls. Negative associations were observed between body appreciation, body dissatisfaction, and disordered eating behaviors in the samples of both genders. Higher body mass index was negatively associated with positive body image in boys as well as girls.

In total, 83.5% of boys and 70.1% of girls declared participation in sports. Scores for the BAS-2-LT were higher in adolescents participating in sports compared with nonparticipating adolescents (for boys 3.48 ± 1.14 vs. 3.04 ± 1.18; for girls 3.32 ± 1.11 vs. 3.06 ± 1.15, respectively, *p* < 0.01).

Furthermore, multiple regression analyses were performed with the other study variables as the predictive and BAS-2 score as the criterion variable separately in boys and girls (Table 6). The results revealed that both models were significant (for boys, F = 34.3; *p* < 0.001; for girls, F = 192.9; *p* < 0.001), explaining 23.3% of the variance of body appreciation variance in boys and 53.6% in girls. Variance inflation factors (VIFs) ranged from 1.0 to 1.4 in boys and from 1.1 to 2.2 in girls, indicating acceptable indices of multicollinearity. The results of linear regression showed that a higher level of body appreciation, self-esteem and a higher ranking in body features representing body functionality for girls were associated with a decreased risk of disordered eating behaviors. On the contrary, higher body mass index and body dissatisfaction predicted disordered eating. For boys, the findings were the same, except for the nonsignificant effect of self-esteem on disordered eating behaviors.

## 4. Discussion

The first aim of the present study was to examine the reliability, validity, and factor structure of the Lithuanian version of the BAS-2 as a screening self-report instrument assessing body appreciation in adolescent samples of both genders. We expected that the BAS-2-LT would be deliberated as a stable test with an adequate internal consistency, and concurrent validity, and it would reflect the original structure. In general, the BAS-2-LT exhibited good psychometric properties. The scale demonstrated high temporal stability of the test–retest reliability over a two-week period, which was good to excellent (ICC range was 0.85–1.00). Furthermore, the internal consistency (Cronbach’s alpha) of the scale was excellent. Thus, the reliability of the scale was fully supported.

Our study confirmed the unidimensional factor structure of the BAS-2-LT in boys and girls, as has been found in previous studies sampling adolescents [24,25]. In line with previous studies, we found measurement invariance between genders. It confirms the sustainability of the scale for measuring body appreciation in adolescent boys and girls in Lithuania. Thus, gender differences found between boys and girls would reflect true attitudinal differences between genders, but not the psychometric differences related to item responses. In the present study, we found that girls demonstrated lower body appreciation, mirroring findings from previous studies [4,19,21,23,24,25]

Furthermore, we aimed to evaluate the convergent validity of the instrument using measures of BMI, body dissatisfaction, disordered eating behaviors, self-esteem, functionality of the body, and participation in sports. In line with the findings of previous studies, we expected that BAS-2 scores would be negatively associated with BMI, body dissatisfaction, and disordered eating, but positively associated with self-esteem, body functionality, and participation in sports in adolescent girls and boys. These assumptions were fully confirmed. As expected, BAS-2-LT scores were negatively associated with BMI, body dissatisfaction, and disordered eating behaviors and positively related to self-esteem and body functionality. These findings are in line with the previous studies that considered adolescent samples [24,25].

As hypothesized, we found the greater body appreciation in sport-participating boys and girls compared with nonparticipating adolescents. These findings might be explained by the assumption that the positive body image is associated with the appreciation of the functionality of body [5]. The associations between body functionality and body appreciation were established in the present study as well. Previous studies in young adults demonstrated that athletes reported higher body appreciation compared with non-athletes [26]. However, studies demonstrated that the quality of motivation (internal motivation) to engage in exercises might be associated with body appreciation [40]. It is important to understand the role of body appreciation in the formation of health-related habits of adolescents in future studies.

Finally, we expected that body appreciation would be associated with the significantly lower levels of disordered eating in both genders, controlling for other study variables. This hypothesis was fully confirmed. In girls and boys, body appreciation together with body functionality and self-esteem were the strongest predictors of lower disordered eating, controlling for BMI and body dissatisfaction. The model for girls explained 54% of variance in disordered eating, and that for boys was 23%. These findings present empirical support for the development of intervention programs that promote positive body image and aim to prevent disordered eating in adolescent boys and girls.

This study provides knowledge about the associations between positive body image and disordered eating behaviors in understudied adolescent populations. Among the strengths of the present study is the solid sample of adolescents of both genders representing cities and rural regions of the country. This study contributes to the growing research into positive body image and adds to the knowledge that BAS-2-LT is a reliable and valid instrument for the measurement of body appreciation in a rarely examined linguistic group of Eastern Europe. The studies of positive body image are of great importance in countries of rapid westernization [16], since lots of young people are experiencing enormous sociocultural pressures to attain beauty ideals and quite low efforts to promote positive body image at schools. Furthermore, the Lithuanian version of BAS-2 might be useful for improving clinical practice.

Beyond its strengths, the present study has some important limitations. The cross-sectional design of the study prevents conclusions about the directions of associations between study variables. Longitudinal studies are recommended to assess the causal associations between positive body image, disordered eating, and health-related lifestyles of adolescents.

## 5. Conclusions

The results of the present study support the psychometric properties of the BAS-2-LT and its use in adolescent Lithuanian samples. Body appreciation is associated with lower levels of disordered eating in adolescent girls and boys. These findings present empirical support for the development in intervention programs that promote positive body image and aim to prevent disordered eating in adolescent boys and girls.

## Figures and Tables

**Figure 1 nutrients-12-00752-f001:**
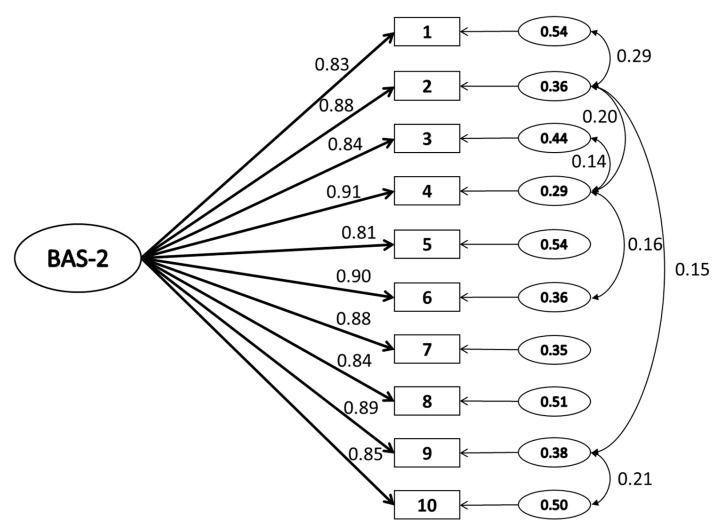
Path diagram and estimates for the one-dimensional model of Body Appreciation Scale-2 (BAS-2-LT) scores from the second split-half adolescent subsample (*n* = 706). Note: The large oval is the latent variable, with the rectangles representing measured variables, and the small circles with numbers representing the residual variables (variances). The standardized path factor loadings are presented (all *p* < 0.001).

**Table 1 nutrients-12-00752-t001:** Descriptive statistics of the Lithuanian Body Appreciation Scale-2 (the adolescent sample, *n* = 1412).

Gender	Mean	Median	SD	Range	Kurtosis	Skewness	Floor (%)	Ceiling (%)
Boys (*n* = 570)	3.41	3.40	1.16	1-5	−0.75	−0.31	6.0	13.2
Girls (*n* = 842)	3.24 *	3.20	1.13	1-5	−0.94	−0.05	3.2	9.3

Note: * *p* = 0.009 as compared with boys.

**Table 2 nutrients-12-00752-t002:** Body Appreciation Scale-2 items in English and Lithuanian and associated item-factor loadings for adolescents from the first split-half subsample (*n* = 706).

Items: English/*Lithuanian*	Boys (*n* = 277)	Girls (*n* = 429)
1. I respect my body/*Aš gerbiu savo kūną*	0.88	0.87
2. I feel good about my body/*Aš jaučiuosi gerai dėl savo kūno*	0.91	0.90
3. I feel that my body has at least some good qualities/*Aš jaučiu, kad mano kūnas turi bent keletą gerų bruožų*	0.88	0.86
4. I take a positive attitude towards my body/*Aš žiūriu į savo kūną pozityviai*	0.91	0.93
5. I am attentive to my body’s needs/*Aš esu dėmesingas savo kūno poreikiams*	0.84	0.83
6. I feel love for my body/*Aš myliu savo kūną*	0.89	0.92
7. I appreciate the different and unique characteristics of my body/*Aš vertinu skirtingus ir unikalius savo kūno bruožus*	0.92	0.88
8. My behavior reveals my positive attitude toward my body; for example, I hold my head high and smile/*Mano elgesys rodo, kad aš pozityviai žiūriu į savo kūną, pvz. aš laikau aukštai iškėlęs (-usi) galvą ir šypsausi*	0.84	0.87
9. I am comfortable in my body/*Aš jaučiuosi patogiai savo kūne*	0.91	0.92
10. I feel like I am beautiful even if I am different from media images of attractive people (e.g., models, actresses/actors)/*Aš jaučiuosi gražus (-i), net jei savo išvaizda skiriuosi nuo medijose rodomų patrauklių žmonių (pvz. modelių, aktorių)*	0.82	0.88
Kaiser–Meier–Olkin test	0.96	0.96
Total variance explained, %	78.1	78.4

**Table 3 nutrients-12-00752-t003:** Lithuanian Body Appreciation Scale-2 confirmatory factor analysis and structural invariance testing across genders from the second split-half adolescent subsample (*n* = 706).

Models	χ^2^/df	GFI	AGFI	TLI	CFI	RMSEA	SRMR
Unconstrained model (general fit across genders)	6.878	0.946	0.897	0.967	0.978	0.065	0.036
Constrained models:							
Measurement weights	6.460	0.941	0.903	0.969	0.977	0.062	0.061
Structural covariance	6.369	0.941	0.904	0.970	0.977	0.062	0.067
Measurement residuals	5.908	0.931	0.910	0.972	0.974	0.059	0.067

Note: χ^2^ = chi-square; df = degrees of freedom; GFI = goodness of fit index; AGFI = adjusted goodness of fit index; CFI = comparative fit index; TLI = Tucker Lewis index; RMSEA = root of the mean square error of approximation, SRMR = standardized root mean square residual.

**Table 4 nutrients-12-00752-t004:** Reliability and validity of the Lithuanian Body Appreciation Scale-2 (the adolescent sample, *n* = 1412).

Gender	Test–Retest Reliability (ICC)	Cronbach’s α	Inter-Item Correlations
Boys (*n* = 570)	0.89	0.97	0.75
Girls (*n* = 842)	0.92	0.97	0.75

Note: ICC = intraclass correlation coefficient.

**Table 5 nutrients-12-00752-t005:** Comparison of means in boys and girls and correlations between body appreciation and additional measures included in the study (the Lithuanian adolescent sample, *n* = 1,412).

Variables	Boys (*n* = 570)		Girls (*n* = 842)		(1)	(2)	(3)	(4)	(5)	(6)
	M	SD	M	SD						
(1) Body appreciation	3.41	1.16	3.24 *	1.13	1	0.28 *	0.59 *	−0.66 *	−0.57 *	−0.28 *
(2) SOQ—body functionality	30.81	4.77	30.48	5.39	−0.026	1	0.21 *	−0.27 *	−0.31 *	−0.13 *
(3) Self-esteem	28.78	5.60	28.39	6.28	0.42 *	0.06	1	−0.46 *	−0.41 *	−0.13 *
(4) Body dissatisfaction	1.17	0.71	1.62 *	0.96	−0.43 *	−0.11 *	−0.29 *	1	0.70 *	0.43 *
(5) Disordered eating behaviors	0.87	0.87	1.72 *	1.33	−0.27 *	−0.11 *	−0.19 *	0.39 *	1	0.39 *
(6) Body mass index	22.00	2.98	20.96 *	3.06	−0.13 *	−0.031	−0.003	0.24 *	0.43 *	1

Note: * *p* < 0.01 as compared with boys for the t test; * *p* < 0.01 for the correlations between study variables; correlations for boys’ subsample are presented under the diagonal; M = mean, SD = standard deviation; SOQ = self-objectification questionnaire.

**Table 6 nutrients-12-00752-t006:** Results of multiple linear regressions predicting disordered eating behaviors in adolescent boys and girls (the adolescent sample, *n* = 1412).

Variables	Boys (*n* = 570)			Girls (*n* = 842)		
	B	β	*P*	B	β	*P*
Body appreciation	−0.074	−0.098	0.025	−0.155	−0.131	<0.001
SOQ—body functionality	−0.014	−0.075	0.046	−0.027	−0.108	<0.001
Self-esteem	−0.009	−0.065	0.116	−0.013	−0.063	0.033
Body dissatisfaction	0.317	0.257	<0.001	0.712	0.511	<0.001
Body mass index	0.075	0.258	<0.001	0.048	0.111	<0.001
F	34.3			192.9		
*P*	<0.001			<0.001		
Model summary	R = 0.48; R^2^ = 0.23			R = 0.73; R^2^ = 0.54		

Note: B = nonstandardized regression coefficient, β = standardized regression coefficient; *p* = level of statistical significance; SOQ = self-objectification questionnaire.

## Data Availability

The dataset generated and analyzed during the current study is not publicly available but is available from the corresponding author on reasonable request.

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
