# Peer review of "Associations between Body Appreciation and Disordered Eating in a Large Sample of Adolescents"

_nutrients, 2020, doi:10.3390/nu12030752_

Round 1
Reviewer 1 Report
This manuscript is an example of well-design and properly conducted examination of the national version of the BAS-2 scale. The strength of the study is well explained methodology from the study design to the data presentation. The text and descriptions are prepared with attention to detail and high awareness, distance and prudence in interpreting obtained results. The manuscript flows well, and the writing style is clear.
Overall, this manuscript is very interesting and adds to the Literature by clearly presenting properties of the Lithuanian version of the BAS-2 scale to use in adolescents samples. However, there is a need to implement minor changes in this manuscript (outlined below).
- Line 116 – Please complete the information on sample representativeness
- Line 125 – the study is part of the larger study – Please write more about the study mentioned
- Line 143-144 – Please add information about the scoring range possible to obtain
- Line 158 - Please add information about the scoring range possible to obtain
- Line 159 – Please add the reference which support the statement “The scales had adequate psychometric (…)”
- Lines 161-172 – Please add the value(s) pf the Cronbach’s alpha(s) for the SOQ as it was reported in descriptions of others scales
- Line 182 - Please add information about the scoring range possible to obtain
- Lines 196-197 – Are the results presented in the Table 5? If so, please complete the information.
- Lines 199-202 – Please add the reference(s) to support the values recommended in the tests (Cronbach’s alpha, Pearson’s correlation).
- Table 6 – Please correct B to β in the 3rd column (Boys)
Author Response
Dear Reviewer,
Thank you for your positive and supportive comments, and for your remarks provided. We made corrections following each comment.
Changes made are highlighted in the manuscript.
This manuscript is an example of well-design and properly conducted examination of the national version of the BAS-2 scale. The strength of the study is well explained methodology from the study design to the data presentation. The text and descriptions are prepared with attention to detail and high awareness, distance and prudence in interpreting obtained results. The manuscript flows well, and the writing style is clear.
Overall, this manuscript is very interesting and adds to the Literature by clearly presenting properties of the Lithuanian version of the BAS-2 scale to use in adolescents samples. However, there is a need to implement minor changes in this manuscript (outlined below).
|
Comments |
Answers |
|
Information completed. |
|
Information extended. |
|
Information added. |
|
Information added. |
|
Reference is provided. |
|
As SOQ scale requires to rank all the ten features from the less important (1) up to the most important (10), different number is attributed to each feature and the Cronbach's α is not calculated. |
|
Information added. |
|
Information completed. |
|
The reference was added. |
|
Corrected. |
Sincerely,
The corresponding author

Reviewer 2 Report
Excellent paper, extending appreciation of positive aspects of self-esteem and body regard in young people, previously studied primarily in negative terms. English is excellent. Statistical analysis looks good, to be confirmed by a reviewer specialized in that area. Only suggestion: many readers would find it useful to have qualitative data - language from study subjects or other volunteers illustrating in their own words the extent of positive feelings captured here in less personal, quantitative terms.
Author Response
Dear Reviewer,
Thank you for your positive and supportive comments. The requested information was added to the introduction section. Changes made are highlighted.
Sincerely,
The corresponding author